# Folding behavior of a T-shaped, ribosome-binding translation enhancer implicated in a wide-spread conformational switch

My-Tra Le[1][*][†], Wojciech K Kasprzak[2][†], Taejin Kim[3], Feng Gao[1], Megan YL Young[1], Xuefeng Yuan[1][‡], Bruce A Shapiro[3], Joonil Seog[4], Anne E Simon[1][*]

[1]Department of Cell Biology and Molecular Genetics, University of Maryland, College Park, United States; [2]Basic Science Program, Leidos Biomedical Research, Inc., Frederick National Laboratory for Cancer Research, Frederick, United States; [3]RNA Biology Laboratory, Center for Cancer Research, National Cancer Institute, Frederick, United States; [4]Department of Materials Science and Engineering, University of Maryland, College Park, United States

**Abstract** Turnip crinkle virus contains a T-shaped, ribosome-binding, translation enhancer (TSS) in its 3'UTR that serves as a hub for interactions throughout the region. The viral RNA-dependent RNA polymerase (RdRp) causes the TSS/surrounding region to undergo a conformational shift postulated to inhibit translation. Using optical tweezers (OT) and steered molecular dynamic simulations (SMD), we found that the unusual stability of pseudoknotted element H4a/$\Psi_3$ required five upstream adenylates, and H4a/$\Psi_3$ was necessary for cooperative association of two other hairpins (H5/H4b) in $Mg^{2+}$. SMD recapitulated the TSS unfolding order in the absence of $Mg^{2+}$, showed dependence of the resistance to pulling on the 3D orientation and gave structural insights into the measured contour lengths of the TSS structure elements. Adenylate mutations eliminated one-site RdRp binding to the 3'UTR, suggesting that RdRp binding to the adenylates disrupts H4a/$\Psi_3$, leading to loss of H5/H4b interaction and promoting a conformational switch interrupting translation and promoting replication.

*For correspondence: my.letra@gmail.com (M-TL); simona@umd.edu (AES)

[†]These authors contributed equally to this work

Present address: [‡]Department of Plant Pathology, College of Plant Protection, Shandong Agricultural University, Tai'an, China

Competing interests: The authors declare that no competing interests exist.

## Introduction

RNAs play key roles in a variety of cellular functions due to their capacity to fold into a number of functionally competent and dynamic structures (*Balvay et al., 2009*; *Dethoff et al., 2012*; *Serganov and Patel, 2007*; *Simon and Miller, 2013*). Driven by conformational plasticity, RNA elements can assume alternative forms required for different biological functions (*Dethoff et al., 2012*; *Simon and Gehrke, 2009*). For example, riboswitches in untranslated regions (UTRs) of bacterial genes can cycle between active and inactive conformations through binding to small metabolites, thereby inhibiting translation or attenuating transcription of downstream genes (*Soukup and Soukup, 2004*; *Winkler and Breaker, 2003*). Regulatory RNA elements found in viral RNA genomes can also switch between mutually exclusive conformations to regulate incompatible processes such as translation and replication (*D'Souza and Summers, 2004*; *Huthoff and Berkhout, 2001*; *Patel et al., 2015*; *Simon and Gehrke, 2009*; *Stammler et al., 2011*; *Stupina et al., 2008*; *Yuan et al., 2009*).

To gain insights into how RNAs transition between alternative conformations, we have studied the folding and unfolding pathways of a 3' cap-independent translation enhancer (3'CITE) found in the 3´ UTR of TCV. TCV is a positive-strand RNA virus (genus Carmovirus, family *Tombusviridae*) with a single uncapped, non-poly(A) genome of 4053 nt encoding five proteins (*Figure 1—figure*

*supplement 1*). The TCV 3'UTR contains at least five hairpins (H4, H4a, H4b, H5 and Pr) and three critical H-type pseudoknots ($\Psi_1$, $\Psi_2$, $\Psi_3$) (*McCormack et al., 2008*; *Simon and Miller, 2013*), a type of RNA structure that forms when loop residues in a hairpin pair with nearby residues to create a second stem-loop. The coaxially stacked double pseudoknots H4a/$\Psi_3$ and H4b/$\Psi_2$ in conjunction with hairpin H5 fold into a T-shaped structure known as the TSS (*McCormack et al., 2008*). The TSS serves as a 3'CITE by binding to 80S ribosomes at the P-site through an interaction with the 60S ribosomal subunit (*Stupina et al., 2008*). Substituting the entire TSS or H4a/$\Psi_3$/H4b with the analogous regions from the related *Cardamine chlorotic fleck virus* (CCFV) led to viral genomic (g)RNA accumulation that was similar to or greater than wild-type (wt) TCV. In contrast, TCV gRNA accumulated poorly when individual hairpins or other combinations of CCFV hairpins and pseudoknots were substituted. These results suggested that all, or a specific subset of hairpins and pseudoknots, function as a structural domain or subdomain (*McCormack et al., 2008*).

In addition to its function in translation, the TSS serves as a scaffold for multiple interactions with surrounding elements extending into the upstream coat protein ORF (*Yuan et al., 2009*, *2012*). The TSS region is also important for replication of a small, non-translated satellite RNA (satC) that contains two TCV-derived 3' regions (*Zhang et al., 2006a*, *2006b*). The basal TSS conformation did not change substantially when bound to ribosomes (*Stupina et al., 2011*), but was significantly altered when a 3'UTR fragment (252 nt) or a 320 nt 3' terminal fragment was bound by the RdRp (*Yuan et al., 2009*, *2012*). Prominent changes included enhanced flexibility of: (1) hairpin H3 in the coat protein ORF and adjacent upstream sequences; (2) the terminal loop of hairpin H4 located just upstream of the TSS; and (3) H4a/$\Psi_3$ in the TSS. Sequences with reduced flexibility included: (1) the 5' side of the basal stem of H5; (2) the sequence between H4b and H5; and (3) the A-rich sequence upstream of H4a. These findings suggested that a mechanism exists for the TSS to transition between conformations required for translation and replication after a threshold level of RdRp was available.

Since the TSS serves as a central hub for interactions with elements throughout the 3'UTR, it seemed likely that disrupting the TSS might trigger the widespread conformational switch that occurs in vitro upon addition of RdRp (*Yuan et al., 2009*, *2012*). For this current report, optical tweezers (OT), a type of single molecule force spectroscopy, and steered molecular dynamic simulations (SMD) were used to examine the folding/unfolding pathways of the TSS. Our results indicate that the TSS follows an unexpected folding pathway that can be reproduced by SMD for conditions that exclude $Mg^{2+}$. In the absence of $Mg^{2+}$, H4b unfolded first, followed by H5 and then H4a/$\Psi_3$, which had been predicted to be the least stable element. In the presence of $Mg^{2+}$, H4b unfolded cooperatively with H5, followed by H4a/$\Psi_3$. $Mg^{2+}$ was not required to form H4a/$\Psi_3$ but enhanced the stability of H4a/$\Psi_3$ and all other elements tested. Disrupting H4a/$\Psi_3$ eliminated the cooperative unfolding of H4b and H5, and H4a/$\Psi_3$ was unstable in the absence of the upstream stretch of five adenylates. Mutating two of these adenylates or adjacent downstream residues eliminated specific binding of RdRp to a 3'UTR fragment, suggesting a model where binding of the RdRp to the five adenylate region disrupts H4a/$\Psi_3$, leading to loss of interaction between H5 and H4b and promoting the conformational switch that interrupts translation and promotes replication.

## Results

### The TSS includes the upstream adjacent adenylates

A 118 nt fragment (positions 3899 to 4016 in TCV gRNA), which includes the TSS along with 15 upstream residues and eight downstream residues, was previously used to analyze ribosome binding to the TSS (*Stupina et al., 2008*) (*Figure 1A*). A truncated version of this fragment that terminated precisely at the two pseudoknots and omitted the upstream five adenylates (5A) (positions 11 to 110 in *Figure 1A*) was subsequently used to determine the TSS structure by NMR and SAXS (*Zuo et al., 2010*). This shorter fragment was not initially stable, and stability required addition of two guanylates at the 5' end that extended $\Psi_3$ and disrupted hairpin H4a. One possibility for why the TSS-only fragment was not stable was the omission of the upstream 5A, as it was subsequently shown that adjacent doublet mutations in 5A (positions 8 and 9 in *Figure 1A*) or single mutations disrupting $\Psi_3$ caused identical enhancements in flexibility of residues throughout the 5A/H4a/$\Psi_3$ region as assayed

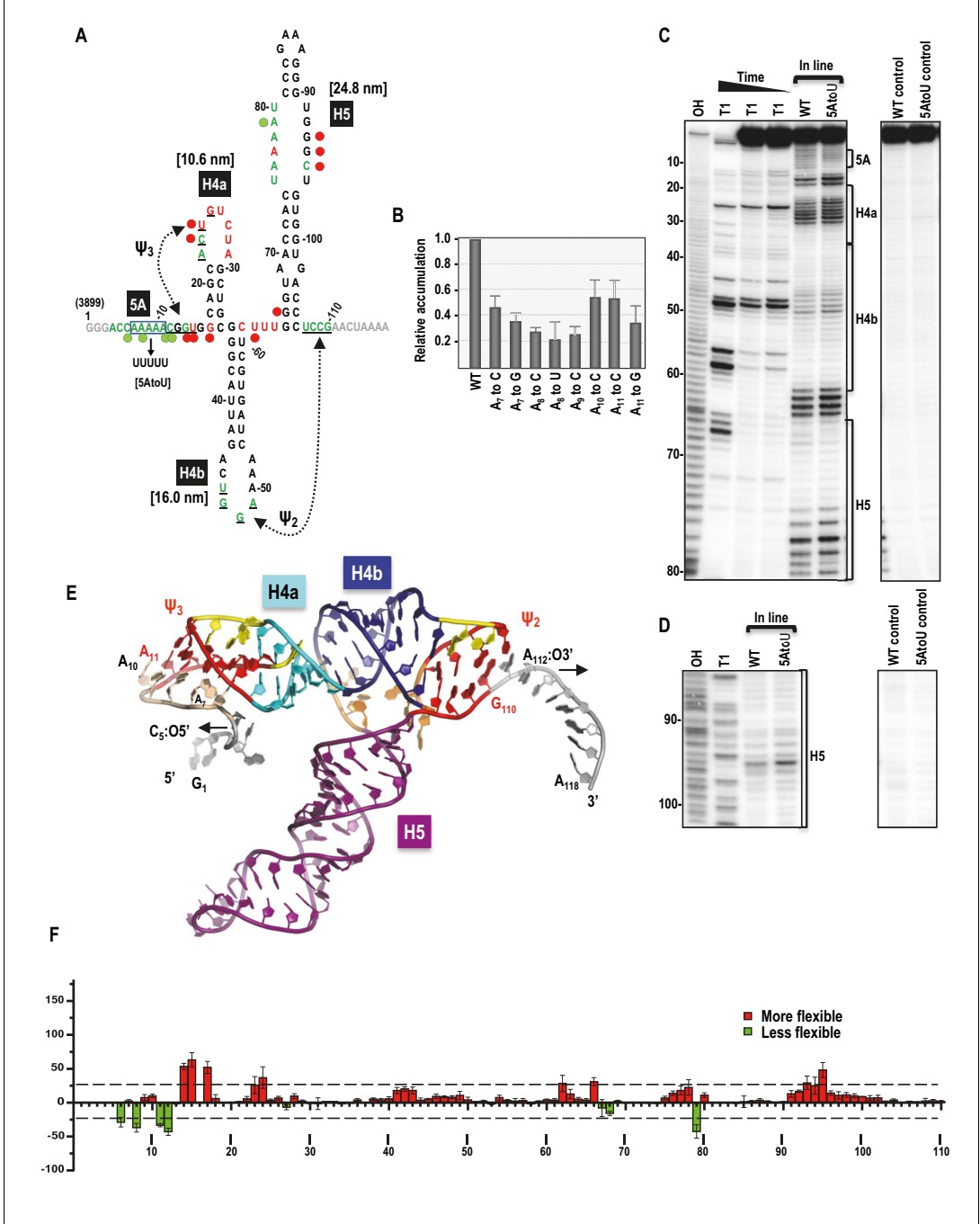

**Figure 1.** Secondary structure, tertiary interactions, and new structural model of the TSS. (A) Sequence of TSS118 near the 3' end of TCV gRNA. Hairpins H4a, H4b and H5 and tertiary interactions $\Psi_2$ and $\Psi_3$ comprise the TSS. Tertiary interactions between underlined residues are indicated by hatched arrows. Predicted contour lengths of hairpins are in brackets. Residues in green and red are moderately and highly susceptible to in-line cleavages, respectively (see C and D). Residues in grey were not evaluated. The 5A region is boxed. Green and red circles denote residues with >25% reduction or enhancement in levels of cleavage, respectively, when 5A is altered to 5U. (B) Accumulation of full-length wt and mutant TCV gRNA in Arabidopsis protoplasts. Individual adenylates were converted to the bases indicated. Data is from three independent experiments and standard deviation is shown. (C) In-line probing of the wt and 5AtoU TSS118 fragments. OH, partial hydroxide cleavage ladder; T1, partial RNase T1 digest of denatured RNA showing the location of guanylates; wt and 5AtoU control lanes were not subjected to in-line cleavage. Intensity of the in-line cleavage products correlates with flexibility of the residues. (D) Longer run of the samples shown in (C). (E) Model of TCV118 corresponding to a 25 ns state in a 40 ns-long MD trajectory. 5' end $C_6$ and $A_7$ through $A_9$ form stable Hoogsteen edge interactions with $\Psi_3$. All 3' end nucleotides beyond $G_{110}$ remain single stranded. A fragment of this model (TSS108), comprising $C_5$ through $A_{112}$, was used as the starting point for all Steered Molecular Dynamics pulling simulations (*Figure 8* and *Figure 8—figure supplement 1*). Nucleotides and atoms where the pulling force was applied ($C_5$:O5' and $A_{112}$:O3')

*Figure 1 continued on next page*

*Figure 1 continued*

were selected to determine the direction of pulling and have the restraints applied to are labeled and arrows indicate the direction of pulling. (**F**) Difference in residue flexibility in wt and 5AtoU fragments. Data is from three in-line probing experiments quantified using SAFA software.

The following figure supplement is available for figure 1:

**Figure supplement 1.** Genome organization of TCV gRNA.

by in-line structure probing (*Yuan et al., 2009*). For this reason, the longer 118 nt fragment (TSS118) that included 5A was used for most of the current study.

Dinucleotide alterations made step-wise throughout 5A were previously shown to be equally detrimental to TCV fitness, with each mutant gRNA accumulating to less than 5% of wt levels in Arabidopsis thaliana protoplasts (*Yuan et al., 2009*). To define the importance of individual adenylates, each was mutated to one or two other nucleotides and mutant gRNAs were assayed for accumulation in protoplasts. Alterations of individual adenylates reduced gRNA levels to between 23% and 55% of wt (*Figure 1B*). Since $A_{11}$ could potentially participate in $\Psi_3$ by pairing with $U_{26}$, it was altered to both a cytidylate, which should disrupt the base-pair and a guanylate, which might maintain the pairing with a G:U pair. $A_{11}$toC was among the least detrimental of the single nucleotide alterations, reducing accumulation by 45% whereas $A_{11}$toG reduced accumulation by 65%. These results suggest that the reduction in gRNA accumulation due to alteration of $A_{11}$ is not connected to its involvement (if any) with $\Psi_3$.

The flexibilities of TSS residues were previously investigated by in-line structure probing using extended fragments that included upstream and downstream hairpins (*McCormack et al., 2008*; *Yuan et al., 2009*). To insure that the structure of the TSS within TSS118 was consistent with its structure in longer fragments, TSS118 was 3' end labelled and the RNA subjected to in-line structure probing. In-line probing reports on the flexibility of individual residues since nucleophilic attack of a ribose 2'OH on the backbone phosphate and ensuing backbone cleavage requires residue flexibility to assume in-line geometry (*Soukup and Breaker, 1999*). As shown in *Figures 1A,C and D*, flexible residues were mainly found in TSS terminal and internal loops. $\Psi_3$, which was originally identified by generating several different compensatory mutations in full-length gRNA, was not resolved by in-line probing of TSS118, similar to what was previously found when subjecting larger fragments to in-line probing and full-length gRNA to SHAPE structure probing (*Chattopadhyay et al., 2015*; *McCormack et al., 2008*; *Yuan et al., 2009*).

To examine the effect of altering 5A on the structure of the TSS, all five adenylates in TSS118 were converted to uridylates (construct 5AtoU) and the mutant RNA fragment subjected to in-line probing. As shown in *Figures 1C and D*, and quantified in *Figure 1F*, selected residues in the 5A region and surrounding sequences of 5AtoU, as well as residues within the H4a loop, displayed flexibility differences of >25% compared with wt. This result is similar to what was found previously when only two adenylates (positions 8 and 9) were mutated within a larger fragment (*Yuan et al., 2009*). In addition, several residues within the large symmetrical loop of H5 and one residue in the linker between H4b and H5 displayed >25% altered flexibility in 5AtoU. Although the 5U in 5AtoU could potentially pair with the five adenylates within the H5 internal loop, only one of the H5 adenylates displayed reduced flexibility, suggesting that structural changes in H5 were not due to an inadvertent pairing opportunity. These results suggest that the TSS forms a similar structure within TSS118 as previously found in longer fragments (*McCormack et al., 2008*; *Yuan et al., 2009*) and that 5A appears to be involved in stabilizing the structure of (at least) H4a/$\Psi_3$.

## Revised structure model of the TCV TSS

The previously published TSS structure model (*McCormack et al., 2008*) corresponded to positions $A_{11}$ through $G_{110}$ in *Figure 1A*. New structure modeling was performed using the previous model as a starting point and extending it at the 5' and 3' ends to include the entire TSS118 fragment. This new model was first subjected to explicit solvent molecular dynamics (MD) simulations, which indicated a potential placement of the 5' end residues $C_6$ and $A_7$ through $A_9$ in the major groove of $\Psi_3$ (*Figure 1E*). The extended 5' end included $A_{11}$ that forms a cis-Watson-Crick/Watson-Crick (cWW)

base pair with $U_{26}$ thus extending and strengthening $\Psi_3$, and added Hoogsteen edge interactions between $A_9$ and $G_{25}/C_{12}$, $A_8$ and $G_{13}$ or $C_{12}$, $A_7$ and $U_{24}$ or $G_{14}$, and $C_6$ and $A_{22}$. These interactions classified with the aid of the DSSR web server (*Lu et al., 2015*) underscore the stability provided by multiple hydrogen bonds and the dynamic nature of interactions between alternative partners. Nucleotides $G_1$ through $C_5$ and $A_{111}$ through $A_{118}$ remained single-stranded.

## Folding and unfolding of the TSS

Genetic analyses and biochemical probing in the presence and absence of RdRp indicated that the TSS and surrounding sequences adopt two substantially different conformations: a basal conformation in the absence of the RdRp that includes the TSS and is likely involved in translation, and an alternative conformation in the presence of the RdRp with a disrupted TSS that is likely used for transcription (*Stupina et al., 2008*; *Yuan et al., 2009*). To investigate this further, we examined the unfolding/folding behavior and stability (i.e., resistance to pulling force) of single TSS molecules under the induction of force using optical tweezers, a form of single molecule force spectroscopy that can be used to investigate conformational changes in single RNAs (*Green et al., 2008*; *Greenleaf et al., 2008*; *Li et al., 2006*; *Liphardt et al., 2001*; *Ritchie and Woodside, 2015*). TSS118 was joined to 500-nt RNA handles at its 5' and 3' ends, which were made double stranded after hybridization to complementary DNA tagged with biotin or digoxigenin (DIG). The biotin DNA handle was attached to a streptavidin-coated polystyrene bead and the DIG DNA handle was attached to an anti-digoxigenin (AD)-coated bead. The streptavidin bead was held in place by a micropipette fixed on a glass chamber and the AD bead was held in an optical trap, which allowed the location of the focal point of the trap to be moved by a piezo (*Figure 2—figure supplement 1*). By moving the focal point of the optical trap mechanically, the AD bead was pulled away from the pipette, allowing the connected single-stranded TSS to unfold. Alternatively, the AD bead in the optical trap was repositioned proximal to the pipette, leading to TSS refolding (*Tinoco et al., 2006*; *Vilfan et al., 2007*). This non-equilibrium measurement is known as force ramping.

Since tertiary interactions like pseudoknots can be dependent on $Mg^{2+}$ (*Lipfert et al., 2014*; *Tinoco et al., 2006*), the TSS unfolding/folding pathways were investigated in the absence and presence of $Mg^{2+}$. In the absence of $Mg^{2+}$, the TSS was expected to unfold with three transitions, corresponding to hairpins H4a, H4b, and H5. At low force (<10 pN), the force- extension curve (FEC) showed monotonic extension of the DNA/RNA handles (*Figure 2A* and *Figure 2—figure supplement 2*), indicating that a single molecule was being stretched. At higher force, three transitions (termed 'rips') labeled '1', '2', and '3' appeared in stepwise fashion, corresponding to successive unfolding of three TSS structural elements (*Figure 2A*, red line). No rips were observed when the force exceeded 20 pN (data not shown). When the force was released, the TSS refolded in three successive steps corresponding to its three unfolding transitions (3, 2, and 1; *Figure 2A*, blue line). Differences in the force required to unfold and fold the individual elements signified hysteric characteristics of the two processes due to the slow unfolding/folding rate of the molecule as compared to the fast loading rate of the applied force, suggesting that these transitions were taking place outside of their thermal equilibrium.

In $Mg^{2+}$, the TSS unfolded with two rips: new large rip (1*) and a small rip (*Figure 2B*), with the small rip appearing identical to rip 3 that occurred in the absence of $Mg^{2+}$. The TSS refolded in $Mg^{2+}$ with three small transitions. Eighty percent of the FECs (N = 64) contained rip 1* when unfolding in the presence of $Mg^{2+}$ while only 14% of the curves (N = 86) contained rip 1* in the absence of $Mg^{2+}$. This suggests that $Mg^{2+}$ enhanced cooperativity among some of the TSS structural elements.

## The order of unfolding of TSS hairpins indicates unusual stability of H4a/$\Psi_3$

Mfold was used to predict $\Delta G$'s of the TSS hairpins (*Zuker, 2003*) and McGenus was used to predict $\Delta G$'s of the pseudoknots (*Bon et al., 2013*). The predicted $\Delta G$'s (in kcals/mole) were H4a: $-6.9$; H4b: $-9.2$; H5: $-21.8$; H4a/$\Psi_3$: $-8.3$; and H4b/$\Psi_2$: $-11.0$. Since the dynamic unfolding behaviors of RNA hairpins and pseudoknots also depend on kinetic barriers to unfolding force, rip assignments required using complementary oligonucleotides to disrupt individual hairpins. In the absence of $Mg^{2+}$, disrupting H4b eliminated rip 1 (*Figure 2C*), and disrupting H5 eliminated rip 2 (*Figure 2D*). In the presence of $Mg^{2+}$, disrupting either H4b or H5 resulted in FECs without rip 1*. Rips 2 and 3

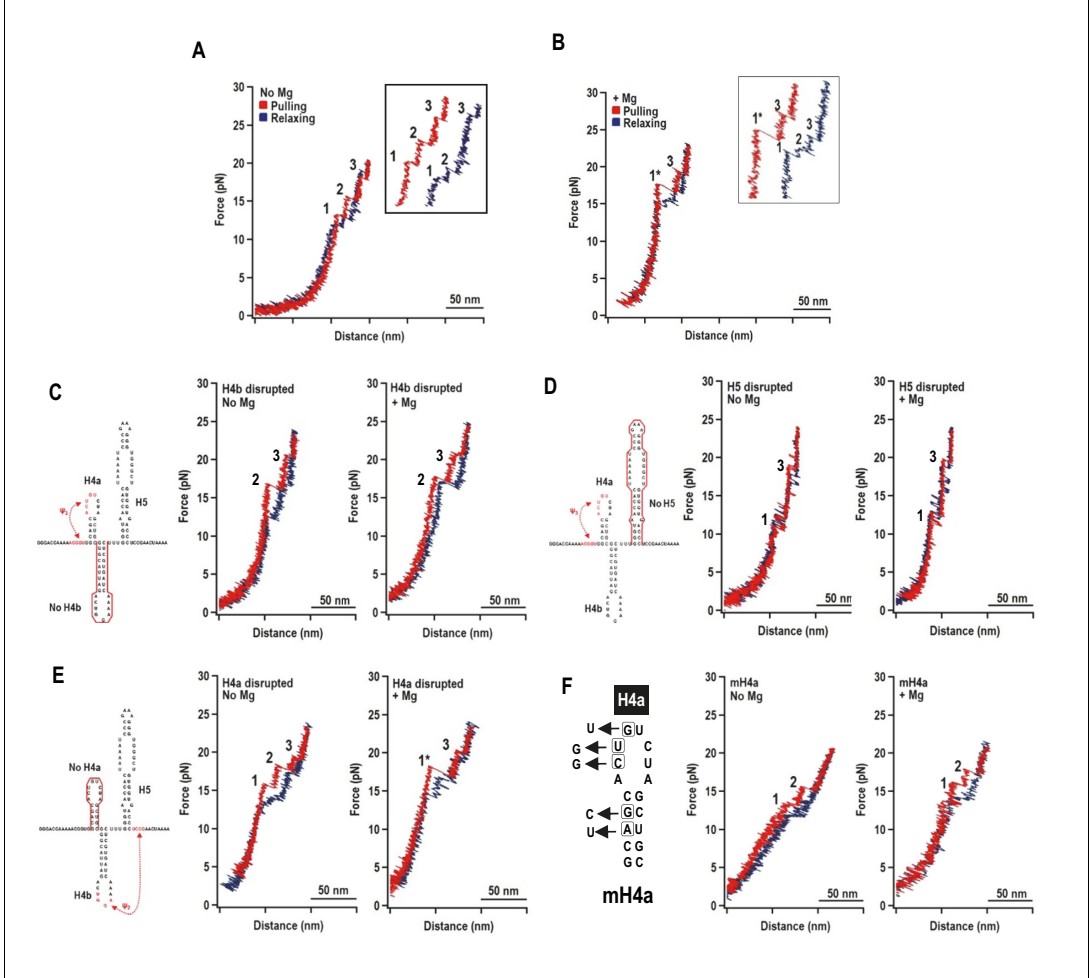

**Figure 2.** Representative force-extension curves (FEC) for the TSS using force ramping. (**A**) FEC of the TSS in the absence of $Mg^{2+}$ (N = 86). The red line represents the unfolding process and blue line represents the refolding process. The three intermediates are labeled. Insets show enlarged transitions. (**B**) FEC of TSS in $Mg^{2+}$ (N = 64). The large and small rips in the unfolding process are labeled 1* and 3, respectively. (**C, D, E**) TSS was annealed with oligonucleotides complementary to H4b, H5, and H4a, respectively. FECs were conducted in the absence (middle) and presence (right) of $Mg^{2+}$. The complimentary oligonucleotides are denoted with a red line. Note that the oligonucleotide complementary to H4a has no effect on the FEC. (**F**) FECs of TSS containing mutations in H4a (mH4a). All rips are numbered as for wt TSS.

The following figure supplements are available for figure 2:

**Figure supplement 1.** Optical tweezer (OT) experimental set-up.

**Figure supplement 2.** Force extension curve (FEC) of the DNA/RNA handles alone.

were present when H4b was disrupted and rips 1 and 3 were present when H5 was disrupted. These results strongly suggest: (1) in the absence of $Mg^{2+}$, rip 1 and rip 2 correspond to unfolding of H4b and H5, respectively; and (2) in the presence of $Mg^{2+}$, H4b and H5 unfold cooperatively as rip 1*, which may also contain $\Psi_2$ that connects H4b with the base of H5.

Addition of the oligonucleotide complementary to H4a had no effect on any rips in the presence and absence of $Mg^{2+}$ (*Figure 2E*), suggesting that the oligonucleotide was blocked from annealing. Therefore, point mutations were introduced into H4a, generating fragment mH4a (*Figure 2F*). When mH4a was subjected to OT in the presence and absence of $Mg^{2+}$, rip 3 (the most stable transition) was eliminated (*Figure 2F*). This suggests that the unusual stability of H4a/$\Psi_3$ (compared to its predicted value) in the presence or absence of $Mg^{2+}$ precluded oligonucleotide annealing. Since rip 3 was present in the absence of $Mg^{2+}$ (*Figure 2A*), $Mg^{2+}$ was not required for H4a/$\Psi_3$ folding, similar

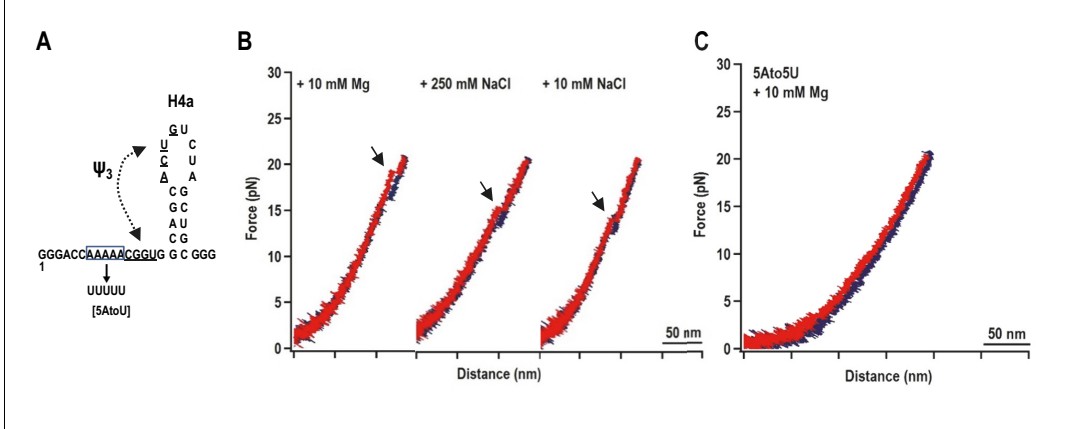

**Figure 3.** H4a/$\Psi_3$ formation during force ramping is not dependent on H4b, H5, or $Mg^{2+}$ and is stabilized by upstream 5A. (**A**) Truncated H4a/$\Psi_3$ fragment. (**B**) Representative FECs in 10 mM $Mg^{2+}$ (N = 22), 250 mM NaCl (N = 72), or 10 mM NaCl (N = 96). Arrows denote H4a/$\Psi_3$ rips. (**C**) FEC (in 10 mM $MgCl_2$) of mutant fragment in which the 5A sequence (boxed) was replaced with 5U (5AtoU).

The following figure supplements are available for figure 3:

**Figure supplement 1.** The unfolding pathway of H4a in the absence of $\Psi_3$.

**Figure supplement 2.** Distribution of the folding force (blue) and unfolding force (red) of H4a/$\Psi_3$ in the fragment shown in *Figure 3A*.

to the ribosome recoding pseudoknot of *Barley western yellows virus* (BWYV) (*Grilley et al., 2006*). In addition, in $Mg^{2+}$, disruption of H4a by point mutations caused rip 1* to apparently be replaced with rips 1 and 2, suggesting that H4a/$\Psi_3$ is required for cooperative association of H4b and H5. These results indicate that: (1) in the absence of $Mg^{2+}$, the TSS unfolds in the order of H4b, H5, and H4a/$\Psi_3$; (2) in the presence of $Mg^{2+}$, the unfolding order is the H4b/H5 complex, followed by H4a/$\Psi_3$; (3) the H4b/H5 complex depends on the presence of H4a/$\Psi_3$; and (4) H4a/$\Psi_3$ is substantially more stable than predicted.

## Effect of 5A on the stability of H4a/$\Psi_3$

When a fragment containing only H4a (positions 13 to 37) was subjected to OT, no rips were observed, suggesting that H4a is not stable in the absence of $\Psi_3$ and/or other surrounding sequences (*Figure 3—figure supplement 1*). To determine if the enhanced stability of H4a/$\Psi_3$ requires additional elements such as 5A, the folding pathway of H4a/$\Psi_3$ was determined using a truncated fragment (positions 1 to 37) composed of H4a/$\Psi_3$, 5A, and additional flanking residues on both ends to reduce disturbance of handles on H4a/$\Psi_3$ structure (*Figure 3A*, left). A single H4a/$\Psi_3$ transition was observed in 10 mM $Mg^{2+}$ near the position of rip 3 (*Figure 3B*, left), suggesting that H4a/$\Psi_3$ can form in the absence of H4b and H5. The critical force required to unfold H4a/$\Psi_3$ was 19.95 ± 0.06 pN (*Figure 3—figure supplement 2*). In the absence of $Mg^{2+}$ and presence of 10 mM or 250 mM NaCl, the single rip was present at forces that were 4 to 5 pN lower (*Figure 3A*, right; *Figure 3—figure supplement 2*), suggesting that $Mg^{2+}$ enhances the stability of the folded structure, similar to what was found for the BWYV pseudoknot (*Grilley et al., 2006*).

Replacing 5A with 5U in the truncated fragment (fragment 5A5U) eliminated the H4a/$\Psi_3$ rip in the presence (*Figure 3C*) and absence (data not shown) of $Mg^{2+}$. Since no other transitions were observed, this result suggests that 5A is required to both form and stabilize H4a/$\Psi_3$. This result supports the in-line probing results showing that mutations in 5A disrupt H4a/$\Psi_3$, as evidenced by enhanced flexibility of residues in the region (*Figure 1*) (*Yuan et al., 2009*).

## The effect of $Mg^{2+}$ on stability and folding kinetics of TSS elements

In addition to non-equilibrium force ramping measurements, holding the force constant, a process known as force clamping, allows RNA hairpins to transition reversibly ('hop') between folded and

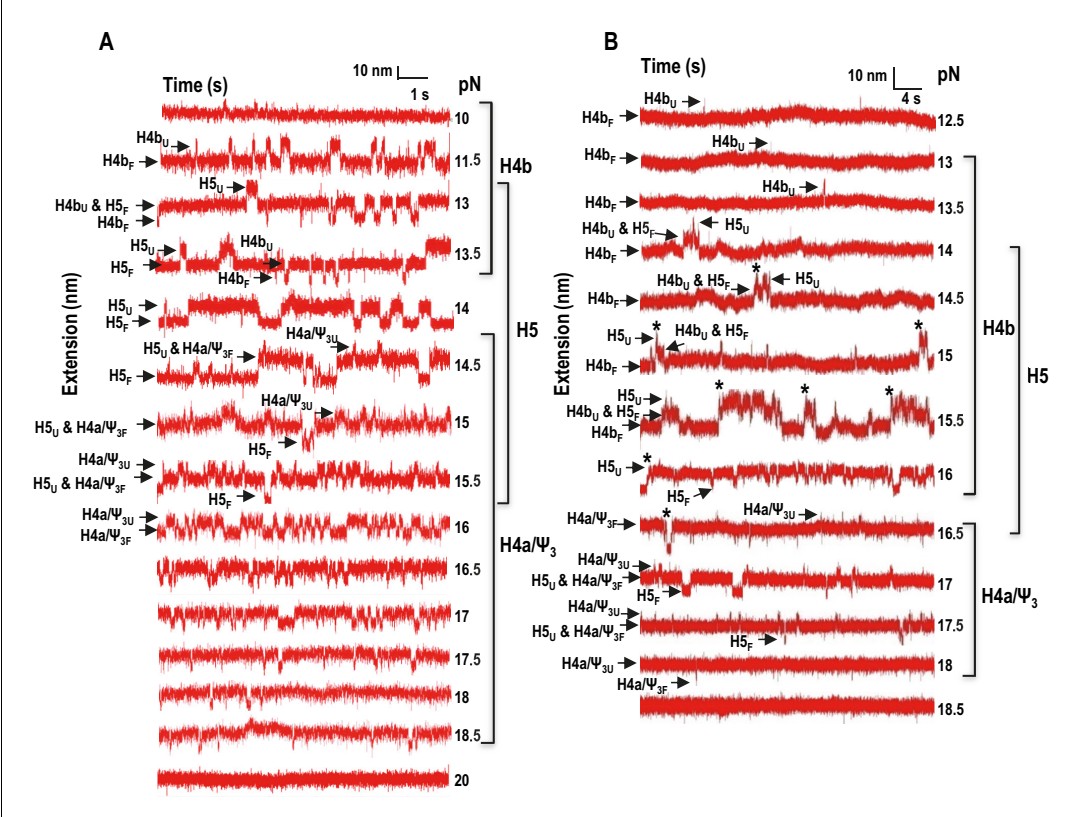

**Figure 4.** Force clamping of the TSS. (A and B) Representative force clamping length versus time traces for TSS118 in 250 mM NaCl and 10 mM $Mg^{2+}$, respectively. The TSS was held at a constant force for between 1.5 and 2 min. The assigned hairpin for each transition is indicated. Subscripts 'U' and 'F' denote unfolded and folded state of the hairpins, respectively, which are indicated by arrows. Brackets to the right denote force ranges of the TSS elements. Asterisks denote cooperative unfolding of H4b and H5.

The following figure supplement is available for figure 4:

**Figure supplement 1.** Measuring the displacement of RNA transitioning between unfolded (U) and folded (F) states using force clamping.

unfolded states at equilibrium, creating lower and upper extensions, respectively, as a function of time (*Tinoco et al., 2006*). The elements involved in each transition in the extension vs time curves were identified by their disappearance followed hybridization with complementary oligonucleotides (for H4b and H5) or mutations in H4a/$\Psi_3$ (data not shown). At critical force levels, in the absence of $Mg^{2+}$, three hopping transitions were observed when TSS118 was held at constant force for 1.5 to 2 min, consistent with the three transitions found by force ramping (H4b, H5 and H4a/$\Psi_3$) (*Figure 4A*). In the absence of $Mg^{2+}$, H4b unfolded at 10 to 13.5 pN, H5 at 13 to 15.5 pN, and H4a/$\Psi_3$ at 14.5 to 18.5 pN (*Figure 4A*). Slowly increasing the force extended the time the hairpins spent in the unfolded state. For example, H4b spent less than 20% of its time in an unfolded state at 11.5 pN, 96% of the time at 13 pN and 100% at 14 pN and higher. $Mg^{2+}$ increased the mechanical stability of all TSS elements, with H4b now unfolding between 13 and 16.5 pN; H5 from 14.5 to 17.5 pN; and H4a/$\Psi_3$ from 16.5 to 18 pN (*Figure 4B*; the slightly higher unfolding force [19.95 ± 0.06 pN] of H4a/$\Psi_3$ obtained by force ramping might be due to the effect of the force loading rate [*Figure 3—figure supplement 2*]). In addition, cooperative unfolding of H4b and H5 was only observed in the presence of $Mg^{2+}$ (hops labeled with asterisks in *Figure 4B* and not observed in *Figure 4A*). $Mg^{2+}$ also caused abrupt changes in the hopping behavior of H4b and H4a/$\Psi_3$. The unfolded state for H4b was less than 1% of the total time at 15 pN, changing to 40% at 15.5 pN, and was nearly 100% at 16 pN (*Figure 4B*). For H4a/$\Psi_3$ in $Mg^{2+}$, the unfolded state was less than 1% of the total time at 17.5 pN and 100% of total time at 18 pN (*Figure 3B*). These results show that unfolding of H4b and H4a/$\Psi_3$

in $Mg^{2+}$ reflects brittle behavior, meaning that in the presence of $Mg^{2+}$, the distance from the unfolded state of these elements is much closer to their transition states.

## Extended lengths of the TSS elements

The estimated contour length of RNA is defined as the length of its maximum possible extension, which is equal to the sum of the length of the component ribonucleotides of the RNA (0.59 nm/ribonucleotide). Experimental contour lengths of the TSS elements were obtained from force clamping experiments, in which the extension histogram of unfolding and folding states of each transition was fitted with Gaussian function. Displacement of RNA transition between unfolding and folding state was converted to contour length using Odjik WLC equation (Equation 2.2, Materials and methods and *Figure 4—figure supplement 1*).

In the absence of $Mg^{2+}$, the contour length of H4b was estimated to be 16.0 nm and the experimental length was 15.3 ± 1.5 nm. The estimated length of H5 was 24.8 nm and the experimental length was 17.0 ± 1.8 nm (H5 had a slightly longer length [~3 nm] at high force; however, this difference was within experimental error). The difference between the estimated and observed lengths of H5 may reflect instability of the H5 lower stem (~5.9 nm) (*McCormack et al., 2008*), which might unfold at a lower force as suggested by the SMD simulations (see below). H4b had a similar average contour length in $Mg^{2+}$ (15.8 ± 3.3 nm) and H5 had a smaller contour length (15.2 ± 2.5 nm) but within the experimental error. These results indicate that, in $Mg^{2+}$, higher force was required to unfold hairpins that retained similar contour lengths, indicating that $Mg^{2+}$ increases the stability of the hairpins.

For $H4a/\Psi_3$, the estimated length was 13.6 nm and the experimental length in the absence of $Mg^{2+}$ was 9.1 ± 1.5 nm. This difference might be due to early opening of some of the $H4a/\Psi_3$ basepairs, as suggested by the SMD simulations (see below). The contour length of $H4a/\Psi_3$ was similar in the presence of $Mg^{2+}$ at all force levels (9.9 ± 3.6 nm) with the exception of 18 pN, where the contour length was 13.5 ± 1.0 nm. These results suggest that, in the presence of $Mg^{2+}$, $H4a/\Psi_3$ may have two kinetics barriers to unfolding force.

## Interactions within the TSS

In $Mg^{2+}$, H4b and H5 exhibited cooperative unfolding in FECs (rip 1*; *Figure 2B*) and had overlapping unfolding forces in force clamping (*Figure 4B*). In addition, when $H4a/\Psi_3$ was eliminated

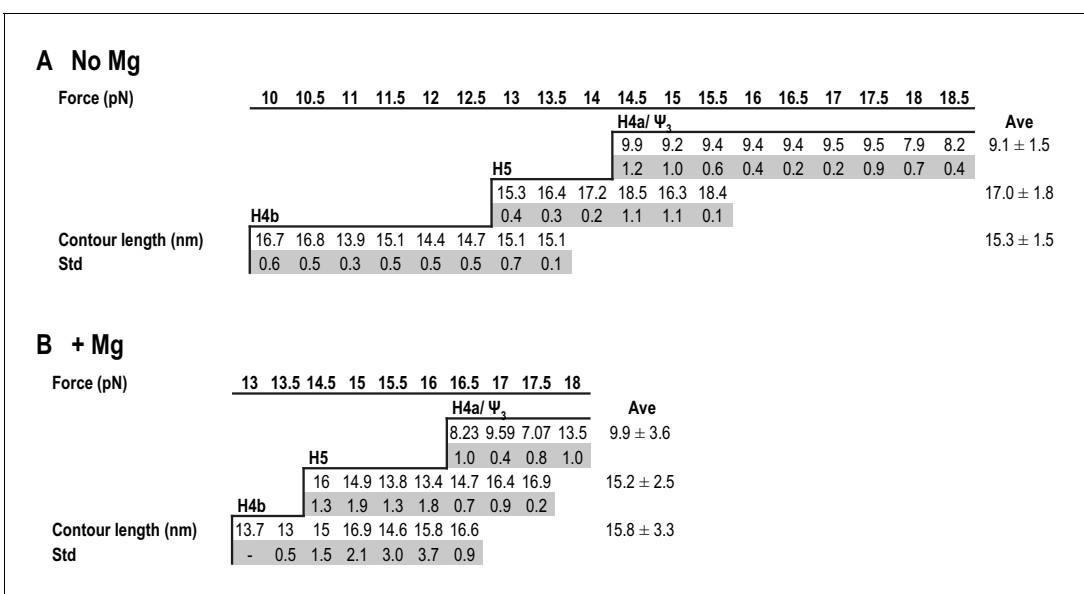

**Figure 5.** Summary of contour lengths of TSS elements obtained by force-clamping. (**A** and **B**) Contour lengths of each transition in 250 mM NaCl and in 10 mM $MgCl_2$, respectively. Std: standard deviation obtained for three independent measurements. Contour length of each transition measured at each force level was averaged (Ave).

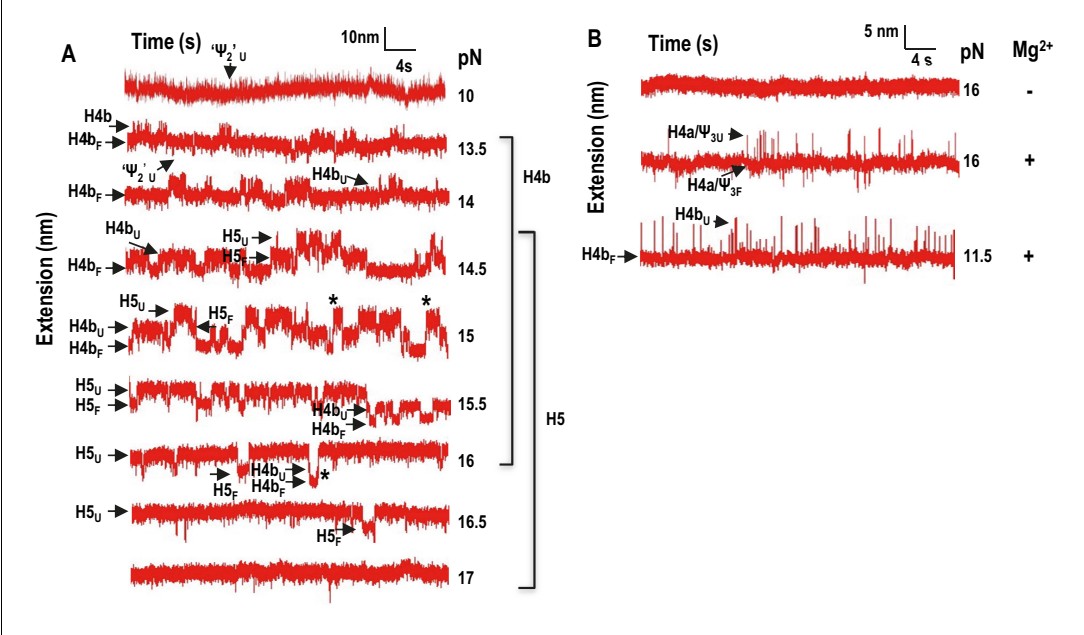

**Figure 6.** Force clamping of TSS with disrupted elements. (**A**) Representative length versus time traces for mutant mH4a in $Mg^{2+}$. H4a and $\Psi_3$ were disrupted by mutations (see **Figure 2F**). '$\Psi_2$' may represent unfolding of $\Psi_2$ and the first few bases of the H4b stem. Asterisks denote cooperative unfolding of H4b and H5. Note that fewer cooperative H4b +H5 unfolding events are observed when compared with wt TSS at the same force level in the presence of $Mg^{2+}$ (see **Figure 4B**). (**B**) Length versus time traces for H5-disrupted TSS. H5-complementary oligonucleotide was annealed to the TSS and force clamping performed using a 5 to 20 pN force range. The hopping behavior of H4b was observed at 11.5 pN in the presence (shown) and absence (not shown) of $Mg^{2+}$. No hopping was observed for H4a/$\Psi_3$ in the absence of $Mg^{2+}$.

(fragment mH4a), rip 1* was replaced with individual H4b and H5 rips, suggesting that H4b and H5 were no longer cooperatively unfolding in the absence of H4a/$\Psi_3$ (**Figure 2F**). A connection between H4a/$\Psi_3$ and H4b/H5 was also evident when fragment mH4a was subjected to force clamping in $Mg^{2+}$ (**Figure 6A**). In the absence of H4a/$\Psi_3$, H4b and H5 transitions were less cooperative and now similar to those found in the absence of $Mg^{2+}$. H4b also unfolded at a lower force range (with multiple hops observed at 13.5 pN compared with 15 pN), and contained a shorter contour length (11.3 ± 1.1 nm compared with 15.8 nm). A new 7.38 ± 0.98 nm transition ('$\Psi_2$') was also observed between 10 and 13.5 pN (**Figure 6A**), which could represent unfolding of $\Psi_2$ and the first few bases of the H4b stem.

To determine if disrupting H4b/H5 affects the stability of H4a/$\Psi_3$ as measured by force clamping, H5 was eliminated by hybridization with its complementary oligonucleotide and the behavior of H4a/$\Psi_3$ was examined. In the absence of $Mg^{2+}$, H4a/$\Psi_3$ hopping was no longer observed (**Figure 6B**), suggesting that, in the absence of $Mg^{2+}$, H5 is needed to stabilize H4a/$\Psi_3$ under force clamping conditions (but not force ramping since rip 3 is still present in H5-disrupted FECs [**Figure 2B**] and when using a fragment containing only 5A/H4a/$\Psi_3$ [**Figure 3B**]). In addition, disrupting H5 caused H4b to start unfolding at a lower force (11.5 pN) in the presence and absence of $Mg^{2+}$ (**Figure 6B**). Altogether, these results suggest that $Mg^{2+}$ stabilizes interactions between H4a/$\Psi_3$ and H4b/H5, and between H4b and H5.

## 5A is implicated in RdRp binding to a 3' end fragment

As described above, the 3' region of the TCV gRNA adopts an alternative conformation upon binding to the RdRp, which significantly affects the structure of the TSS and other nearby elements (**Yuan et al., 2010**, **2009**). Since the TSS is a central hub for RNA interactions within the 3'UTR and upstream sequences (**Yuan et al., 2012**), it seemed possible that the widespread conformational rearrangement could be triggered by RdRp binding to, and disrupting, the TSS. One consequence of RdRp binding to fragments containing the TSS was reduced flexibility of 5A residues, suggesting

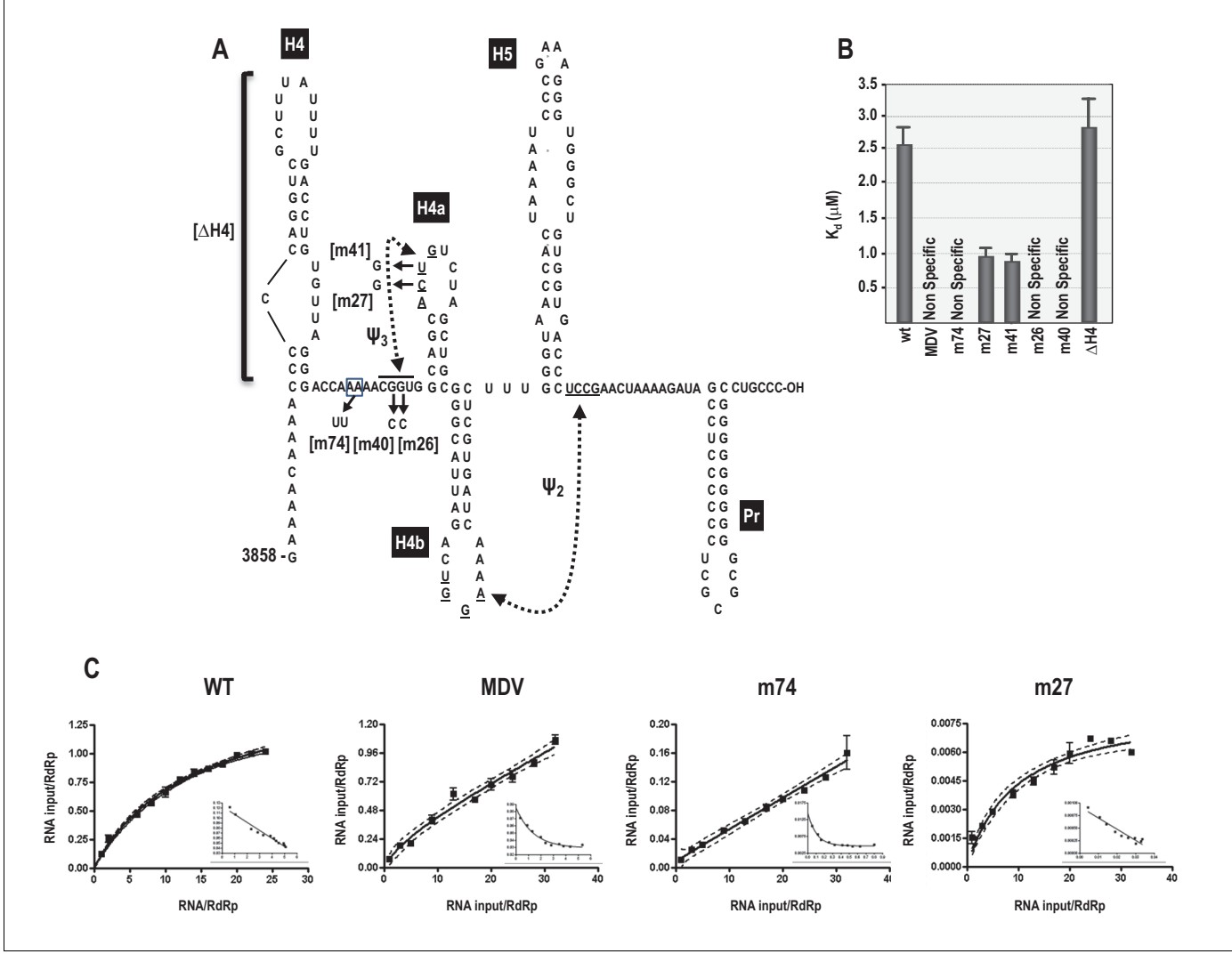

**Figure 7.** RdRp binding to mutant and wt 3' fragments in vitro. (**A**) Sequence and structure of the fragment used for RdRp binding. Locations and names of alterations are shown. ΔH4 is a complete deletion of hairpin H4. Numbering is from the 5' end of the viral genomic RNA. (**B**) Five to 160 pmol of labeled wt and mutant fragments or control MDV RNA was combined with 11.4 pmol of purified, recombinant TCV RdRp, and bound RNA detected following filtration. MDV is a subviral RNA associated with Qβ bacteriophage and serves as a negative control for RdRp binding. (**C**) Selected saturation binding curves are shown together with Scatchard plots (insets). The similarity in non-linear Scatchard plots between MDV and m74 is interpreted as non-specific. For all assays, $K_d$ were calculated from three independent experiments conducted in triplicate. Standard error bars are shown.

that this motif might be a target site for the RdRp (*Yuan et al., 2009*). To assess the validity of this hypothesis, RdRp filter binding assays were conducted on an extended 3' fragment (positions 3858 to 4053) to determine if binding dissociation constants were affected by mutations in 5A. Recombinant TCV RdRp bound to the wt fragment with a $K_d$ = 2.6 μM (*Figure 7*). In contrast, RdRp did not exhibit specific one-site binding to MDV (midivariant RNA, 221 nt), which is a non-template satellite RNA associated with Qβ bacteriophage. A two nucleotide mutation in 5A at positions 8 and 9 (A to U; m74) prevented specific binding of RdRp to the fragment. Loss of specific binding supports our previous finding that m74 also reduced RdRp-directed in vitro transcription of this fragment by 3.6-fold (*Yuan et al., 2009*). The ability of the RdRp to continue transcribing low levels of complementary strands in the absence of 5A binding is likely due to the presence of the 3' terminal Pr hairpin, which in combination with upstream hairpin H4 can promote a low level of transcription (*Sun and Simon, 2006*). Since mutations in 5A disrupt the stability of H4a/Ψ₃, RNA fragments were also examined that contained mutations in Ψ₃. Two individual alterations in the loop of H4a (m27, m41)

improved binding of the RdRp to the template by 2.5-fold. In contrast, mutations in the two partner residues of m27 and m41 located just downstream of 5A (m26, m40) eliminated one-site binding to the template (*Figure 7*). These results suggest that disruption of H4a/$\Psi_3$ (which occurs when the adenylates are altered in m74) is not the proximal cause of the loss of specific RdRp binding, but rather the RdRp may be specifically interacting with 5A and adjacent downstream sequences, and this interaction may be facilitated by the absence of $\Psi_3$.

Since specific binding was lost when residues just downstream of the 5A sequence were altered, we investigated whether binding might extend into upstream hairpin H4, which is known to be critical for TCV accumulation (*Yuan et al., 2009*, *2012*). Deletion of H4 ($\Delta$H4) had no significant effect on RdRp binding, suggesting that the binding region may be confined to the sequence between H4 and H4a.

## Molecular dynamics simulations of TSS unfolding

OT provides a coarse level view of unfolding events whereas equivalent SMD simulations have the potential of revealing subtle interactions at the atomic level. However, before simulations can provide fine grain information, the simulated order of unfolding of the TSS helices must agree with the experimental OT results. OT is performed at pulling speeds several orders of magnitude slower than practical for simulations as the total time required to perform one simulation at the experimental pulling speed would be prohibitive. Therefore, we first assessed the impact of pulling speed over a range of two orders of magnitude on the unfolding results simulated with SMD.

Explicit solvent MD simulations of nucleic acids are considered to be more accurate as all water/salt atoms around the solute are accounted for in the simulation. However, the full extension of the TSS model required a solvent box approximately 700 Å long, resulting in a large simulated system (up-to ~480,000 atoms) and limiting the slowest pulling speed to 0.05 Å/ps, (requiring three weeks to run one simulation). Therefore, computationally much faster MD method using the implicit solvent Generalized Born model (GB) was employed to explore an order of magnitude slower simulated pulling speeds. Both types of simulations used explicit or simulated concentration of monovalent $Na^+$/$Cl^-$ counter ions. $Mg^{2+}$ ions were not included as their initial placement is unknown. Furthermore, the additional level of complexity introduced by the SMD protocol would not allow sufficient time for the potential coordinating effects of $Mg^{2+}$ to be established, and thus their inclusion was deemed insufficiently accurate for use in this study. Therefore, all simulation results are based on NaCl and are compared to experimental conditions in the absence of $Mg^{2+}$.

Initial explicit solvent SMD simulations performed on fragment TSS108 ($C_5$ through $A_{112}$) at pulling speeds of 1.0 Å/ps and 0.5 Å/ps terminated with H5 and H4b helices remaining partly base-paired, indicating that the RNA chain was ahead of its approximate equilibrium state. Full base-pair opening for the extension of the 5′ and 3′ ends could be achieved by an additional equilibration step at the end of these simulations (i.e., with holding to maintain the extension, but without pulling) or by slowing down the pulling speed to 0.1 Å/ps. These runs yielded an unfolding sequence different from the experimental results, even at the 0.05 Å/ps pulling speed (resulting in the longest practical explicit solvent simulation times). The key observation based on these simulations was that pulling speed affected the orientation of helices relative to the direction of pulling (refer to Appendix 1 for more details). This led to formation of new, transient tertiary interactions by H4b after all hydrogen bonds between H4b and $\Psi_2$ opened, increasing the resistance to pulling and impacting the order of TSS unfolding. These observations are consistent with results of OT unfolding of a pseudoknot (*Chen et al., 2007*). Introducing a pause in the pulling simulation immediately after $\Psi_2$ opening to re-equilibrate the system and avoid transient H4b interactions changed the order of TSS opening events to H4b opening first, followed by H5 and then H4a.

Given this qualitative agreement between the experimental and computational results from the 'paused' (locally slowed-down) simulation in explicit solvent, we switched to the implicit solvent (GB) protocol that allowed for an order of magnitude decrease in pulling speed. Three variants of the GB protocol (see Materials and methods) yielded generally consistent results. In 70% to 100% of the SMD simulations (depending on the parameters used) with a constant pulling speed of 0.01 Å/ps, H4a/$\Psi_3$ opened last, and in 40% to 50% of all the simulations, the full sequence of events was in agreement with the experimental OT results (H4b opening first, H5 second, and H4a/$\Psi_3$ last). Implicit solvent simulations at 0.005 Å/ps pulling speed yielded comparable results. The

demonstrated applicability of the implicit solvent MD to optical tweezers pulling simulations of highly structureed nucleic acids is novel in its own right, and worth noting as such.

Consistent with the slowest 'paused' explicit solvent simulation (refer to Appendix 1 for more details), implicit solvent simulations that followed the OT order of TSS unfolding showed none or only minor transient interactions between the H4b terminal loop and other tertiary interaction partners. In addition, as was the case in the majority of the explicit solvent simulations at higher pulling speeds, in the few implicit solvent simulations in which H4b opened last, strong tertiary interactions occurred between H4b loop residues and other nucleotides after the original base pairs involved in $\Psi_2$ opened. The resulting tertiary structure was oriented approximately in parallel with the direction of pulling. Even though topologically it is a pseudoknot, at the 5' end of the opening H4b was pulled through a loop (as if it was a true knot in 3D; *Figure 8—figure supplement 1*) that increased the number of opportunistic interactions and resulted in more resistance to pulling than H4a/$\Psi_3$, which is not 'knotted'. In the absence of a new pseudoknot replacing the $\Psi_2$ interactions with H4b, H4a/$\Psi_3$ offered the most resistance to pulling and opened last. The parallel orientation of H4a/$\Psi_3$ with respect to the direction of pulling increased its stability beyond what the free energy calculations (Mfold and McGenus) would suggest, again, in agreement with another OT pseudoknot unfolding study (*Chen et al., 2007*).

A sample SMD-simulated sequence of unfolding events is illustrated in *Figure 8* and shown in *Video 1* the Supplementary SMD trajectory movie. Unfolding begins with $C_6$ and 5A adenylates losing their initial triple interactions with $\Psi_3$ and the extended pairing in $\Psi_3$ ($A_{11}$). Next, the stacking between H4a and H4b is opened ($C_{34}$ and $G_{35}$ are pulled apart). This is followed by $\Psi_2$ opening and H4b pivoting toward vertical (perpendicular to the direction of pulling) without forming any transitional tertiary interactions and unwinding fully as the first of the major TCV TSS helices. In the next stage, the entire H5 opens. Finally, H4a/$\Psi_3$ opens with H4a helix pairs opening from the 5'−3' end toward the hairpin. The illustrated simulation exhibited separation of H4b from pseudoknot $\Psi_2$ without any transient tertiary interactions, and this clean transition appears to contribute to the clearly defined sequence of events. The individual helices open sequentially, one at a time, rather than partially and in parallel (concurrently). The latter scenario was observed in many SMD simulations at all pulling speeds, although it was most pronounced at faster pulling speeds. For example, rather than 'waiting for its turn' to begin opening, H4a/$\Psi_3$ opens its 5'−3' base pair or pairs and pauses as weaker base pairs open elsewhere in the TCV. Gradual changes in the hydrogen-bond interactions that maintain the pseudoknotted 3D topology of the 5' end are similar to the case of H4b 'slipping' (see Appendix 1). This scenario could explain the difference between estimated (13.6 nm) and experimental (9.1 ± 1.5 nm) contour length of H4a/$\Psi_3$, which would not include any 'premature' base pair openings. In multiple runs, we observed partial opening of H4a/$\Psi_3$ foreshortening the contour length, first to 11.2 nm ($C_{12}$ through $G_{30}$ or $G_{13}$ through $C_{31}$), and then to 9.4 nm ($U_{15}$ through $G_{30}$), after which the remaining structure opened rapidly. Similar foreshortening of the contour length was also observed for the opening simulations of H5. A pause in unzipping is predicted for the substructure between $C_{71}$ and $G_{100}$, followed by a rapid opening of H5, especially across the non-canonical pairing in its large internal loop. The contour length of this substructure is 17.7 nm, well within the experimental measurements. The experimentally obtained FECs indicate rapid opening (unzipping) of entire helices, one helix at a time, while our simulations include both scenarios and suggest that partial helix opening may play a role in the process.

## Discussion

The TSS is a dynamic element that adopts two biologically relevant conformations in vitro; a highly stable T-shaped structure (*McCormack et al., 2008*) and a substantially altered conformation that appears upon RdRp binding, and reverts back when the RdRp is degraded (*Yuan et al., 2009*). How the TSS along with at least 300 nt at the 3' end of the TCV gRNA convert between the two conformations is unknown. In the T-shaped structure, the TSS contains hairpin H5 in conjunction with the coaxially stacked pseudoknots H4a/$\Psi_3$ and H4b/$\Psi_2$ (*McCormack et al., 2008*). Based on Mfold-predicted ΔG's for the three TSS hairpins, H4a (in the absence of the pseudoknot) should be the least stable element (ΔG = −6.9 kcal/mol), followed by H4b (ΔG = −9.2 kcal/mol), and H5 (ΔG = −21.8 kcal/mol). In the absence of $Mg^{2+}$, H4b unfolded experimentally between 10 and 13.5 pN, and H5 unfolded between 13 and 15.5 pN (*Figures 2* and *4*). In addition to H5 being substantially less stable

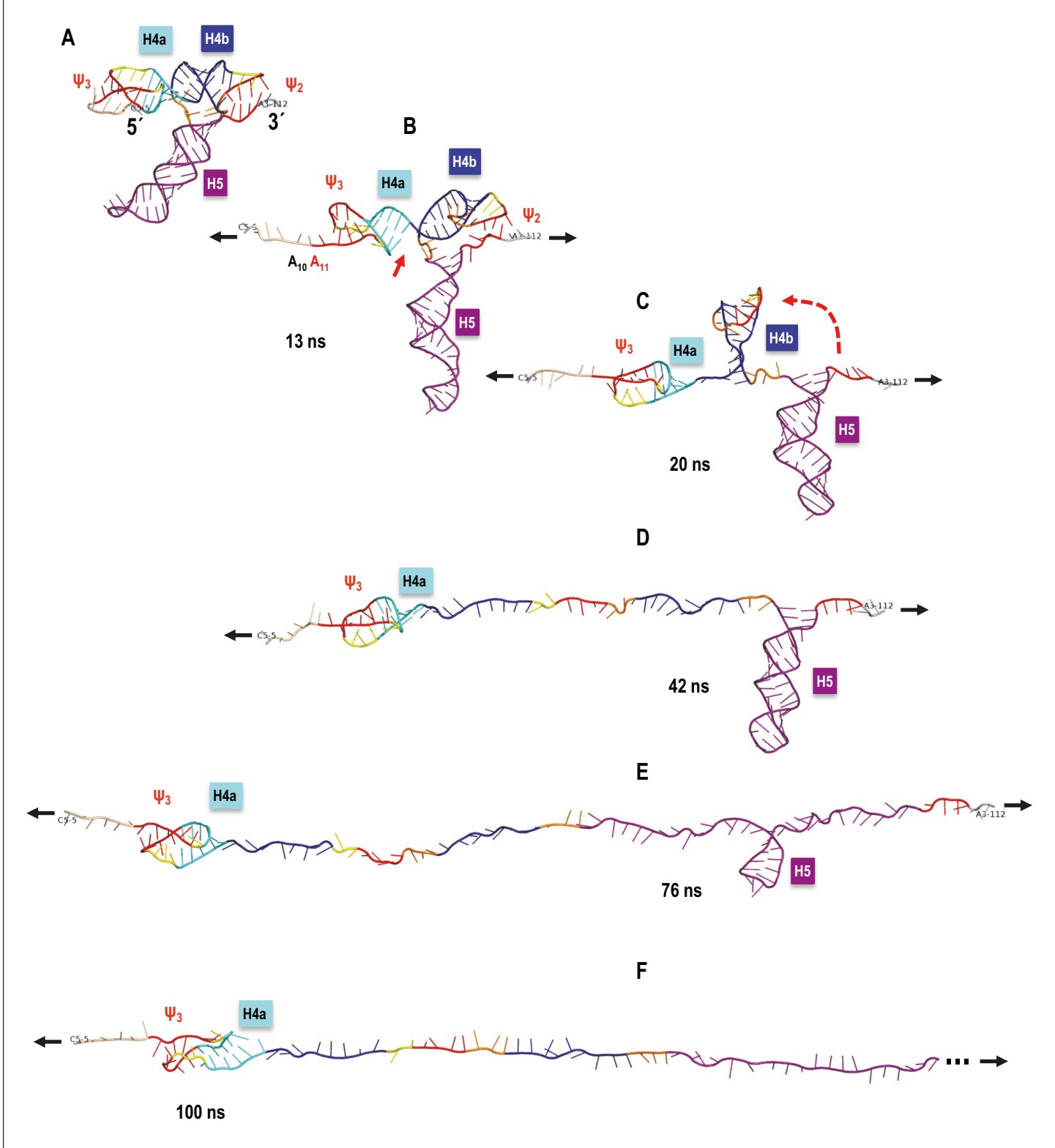

**Figure 8.** TCV TSS pulling simulation with Steered Molecular Dynamics (SMD). Snapshots from an implicit solvent SMD simulations with a pulling speed of 0.005 Å/ps for the TSS fragment between $C_5$ (labeled as 5′) and $A_{112}$ (labeled as 3′), inclusively. (A) Initial TSS model with the 5As held by triple base interactions in the major groove of $\Psi_3$. The initial distance between the ends is 66 Å. (B) After 13 ns of SMD simulations (~130 Å extension) the 5A are open and the initial end-to-end stacking between H4a and H4b also opens. (C) After 20 ns (~167 Å extension) H4b pivots to the vertical orientation shown, following opening of $\Psi_2$ at approximately 19 ns time point. (D) After 42 ns (~279.5 Å extension) H4b is fully pulled apart, while H4a and H5

*Figure 8 continued on next page*

*Figure 8 continued*

remain fully base-paired and $\Psi_3$ nucleotides maintain a pseudoknot. (E) After 76 ns (~419 Å extension) H4a/$\Psi_3$ holds while H5 is opening gradually with a pronounced pause at the C71-G100 base pair (refer to the text for details). (F) After H5 opens fully around 85 ns of the simulation, the remaining 5´ end of the TSS (H4a/$\Psi_3$) begins to open around the 100 ns point shown (~566 Å of full extension) and is free of any base pairs after reaching the extension of 612 Å after approximately 109 ns (not shown). The 3´ end of the TSS in (F) is truncated (indicated by the three dots) to maintain the same scale in all panels.

The following figure supplement is available for figure 8:

**Figure supplement 1.** TSS simulated unfolding pathway showing the impact of transitional tertiary interactions between H4b and the unfolding chain on the coaxial orientation of H4b relative to the direction of pulling resulting in an altered order of unfolding events.

than expected, it also has a smaller than expected contour length (15 to 17 nm compared with the expected length of 25 nm). Reduced stability of H5 may reflect instability of its lower stem (~5.9 nm) (*McCormack et al., 2008*), which might unfold at a lower force range, as suggested by SMD. In the absence of the lower stem, the $\Delta G$ of H5 is predicted to be −14.9 kcal/mol.

No rip corresponding to H4a alone was found, either in TSS118 (*Figure 2*) or when alone in a truncated fragment (*Figure 3—figure supplement 1*), suggesting either that H4a does not form independently or that its unfolding is obscured by the unfolding of the DNA/RNA handles. In the presence of upstream sequences, including 5A and sequences that form $\Psi_3$, H4a/$\Psi_3$ is the most stable TSS element and folding is independent of Mg$^{2+}$ (*Figures 2* and *3*). High force was also necessary to unfold the human telomerase pseudoknot (*Chen et al., 2007*). Stability of H4a/$\Psi_3$ was dependent upon association with the upstream 5A as the element did not form when the adenylates were converted to uridylates (*Figure 3C*). 5AtoU also caused structural changes in H4a/$\Psi_3$ residues (*Figure 1*), similar to what was previously found when just $A_9$ and $A_{10}$ were altered, or $\Psi_3$ was disrupted (*Yuan et al., 2009*). The importance of 5A for TSS structure is consistent with previous findings that 5AtoU decreased ribosome binding and translational efficiency in luciferase reporter constructs (*Stupina et al., 2008*).

In the new TSS model (*Figure 1E*), A7 through A10 help to support $\Psi_3$, which is extended to include the $A_{11}$–$U_{26}$ base pair, but the triple interactions that they form did not offer much resistance to pulling. It is therefore possible that interactions other than those proposed in the model and that are stable in MD simulations form between 5A and H4a/$\Psi_3$. H4a/$\Psi_3$ may also be highly stable due to its coaxially stacked stems, creating base-pair planes perpendicular to the direction of the unfolding force requiring high force to overcome resistance of multiple base pairs at the same time. Computer simulations of the unfolding pathway support this hypothesis by indicating that the structural resistance to pulling is increased for the TSS pseudoknotted motifs beyond their free energy predictions by their orientation parallel (and nearly co-axial) to the direction of pulling (along the top of the TSS's 'T' shape). Of the two pseudoknots, the long-distance $\Psi_2$ is weaker than the short-distance H-type $\Psi_3$ and opens first, exposing H4b and H5, now oriented approximately perpendicular to the direction of pulling. H4b is shorter (and has a higher predicted free energy) and opens first, followed by H5. Beside the alignment between the motif axis and the direction of pulling, akin to 'static resistance', we observed what could be labeled as 'kinetic resistance' based on the formation of transient ('slipping') tertiary interaction between the helices oriented (initially or transiently) along the direction of pulling and in the proximity of other (downstream or upstream) parts of the TSS. For example, $\Psi_3$ opens gradually and, as the entire $\Psi_3$ motif opens, transient tertiary interactions outside of the original $\Psi_3$ base pairs maintain the pseudoknot topology, thus increasing the resistance to pulling and delaying the opening of H4a/$\Psi_3$.

Mg$^{2+}$ is important for stabilizing RNA tertiary structures (*Draper, 2004*; *Pyle, 2002*; *Tan and Chen, 2011*; *Woodson, 2005*) and has been shown to be a key factor in controlling conformational switches in a number of RNAs (*Choudhary and Sigel, 2014*; *Reining et al., 2013*; *Suddala et al., 2015*). In Mg$^{2+}$, cooperative association of H5 and H4b led to a new large rip in the FECs (1*), which unfolded at a lower force than H4a/$\Psi_3$ (*Figure 2B*). In addition, stability of all three TSS transitions increased by 2 to 3 pN over the force required in the absence of Mg$^{2+}$ (*Figures 4* and *6*), and fully stretched H4a/$\Psi_3$ (13 nm) was only obtained in Mg$^{2+}$ at high force ($\geq$18 pN) (*Figure 5*). These results imply that Mg$^{2+}$ increases the kinetic barriers to the mechanical unfolding of the TSS elements. In

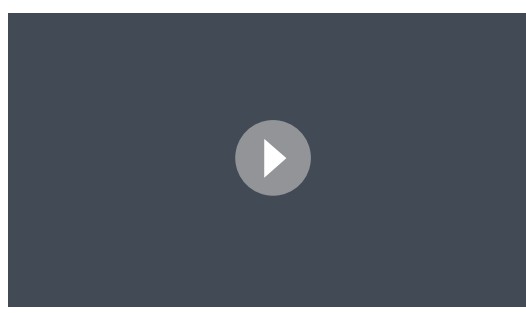

**Video 1.** TCV TSS optical tweezers pulling simulation movie. A movie of aTCV TSS pulling simulation with Steered Molecular Dynamics (SMD). This movie shows the entire SMD trajectory (sampled to be viewed in a minute) for a simulated pulling speed of 0.005 Å/ps for the TSS fragment between $C_5$ (labeled as 5′) and $A_{112}$ (labeled as 3′), inclusively. This is a full trajectory from which the key state snapshots are shown in *Figure 8*. The initial distance between the ends is 66 Å. At ~130 Å of extension the 5′ 5A open, followed by $\Psi_2$ opening (~167 Å), full H4b (~279.5 Å), H5 (with a pause at the C71-G100 base pair) opening fully at ~494 Å. Finally, the TSS's 5′ end (H4a/$\Psi_3$) opens between ~566 Å and 612 Å of the full molecule extension. The simulation stops at 700 Å end-to-end extension.

addition, the hopping behavior of H4b and H4a/$\Psi_3$ in $Mg^{2+}$ changed abruptly between folded and unfolded states between 15.5 and 16 pN for H4b and H5, and between 17.5 pN and 18 pN for H4a/$\Psi_3$ (*Figure 4*), indicating that the presence of $Mg^{2+}$ increased the stability of these elements and suppressed their unfolding. This could explain the refolding dynamics of the TSS, which experimentally assumes a conformation compatible with ribosome binding in the absence of the RdRp (*Yuan et al., 2009*).

There are several lines of evidence indicating that H4b and H5 associate as a unit within the TSS in $Mg^{2+}$. For example, disruption of either H4b or H5 led to the disappearance of rip 1* (*Figures 2C and D*), indicating cooperative unfolding of the two hairpins. Furthermore, disruption of H5 caused H4b to unfold at 11.5 pN in the presence of $Mg^{2+}$ (*Figure 6B*), which was the same force level as in the absence of $Mg^{2+}$. In addition, alterations that disrupted H4a/$\Psi_3$ (mH4a) converted rip 1* to the individual H4b and H5 rips (rips 1 and 2; *Figure 2F*), indicating that H4b/H5 cooperation required H4a/$\Psi_3$. A connection between H5 and H4b was also previously suggested by finding second-site mutations in H4b when the lower bulge loop of H5 was mutated (*McCormack, 2007*). Although FECs indicated that H4a/$\Psi_3$ was stable in the absence of H5 or H4b (*Figures 2C and D* and *Figure 3*), disruption of H5 eliminated the normal H4a/$\Psi_3$ transition in the absence of $Mg^{2+}$ (*Figure 6B*).

Our finding that RdRp binding to a 3′ fragment is disrupted when 5A is mutated suggests a new model for how the TSS and surrounding sequences transition between alternative conformations. We propose that a basal conformation of the 3′UTR contains the TSS, which serves as a 3′CITE for translation of the RdRp from the gRNA template when the gRNA first enters a cell. When a threshold level of RdRp has been synthesized, the RdRp binds to the 5A region that includes $\Psi_3$ residues, disrupting H4a/$\Psi_3$, which consequently disrupts the interaction between H4b and H5. Since the TSS serves as a central hub for interactions throughout the region (*Yuan et al., 2012*), disruption of the TSS would likely cause additional structural changes leading to the alternative conformation of the RNA that is found upon RdRp binding. Although this site of RdRp binding is distal to the 3′end where transcription commences, previous results indicated that a structural connection exists between the 3′ terminus and the 5A region (*Yuan et al., 2012*), which could place the 3′ end in a position to be accessed by the 5A-bound RdRp. We believe that this scenario provides an intriguing blueprint for possible RNA structural transitions in other viruses with 3′ translation elements.

## Materials and methods

### In-line structure probing
DNA fragments coding for wt and mutant TSS were amplified by polymerase chain reaction (PCR) using plasmids containing the full-length TCV gRNA. TSS mutants were generated by site-directed mutagenesis using two complementary primers and high fidelity pfu polymerase (NEB) following a protocol previously described (*Zheng et al., 2004*).

RNA transcripts were synthesized in vitro using T7 promoter-containing PCR fragments as templates and then purified by electrophoresis through 1.5% agarose gels. Purified RNA transcripts were 3′-end labeled with [α-$^{32}$P] dATP as previously described (*Huang and Szostak, 1996*). Briefly, a 17-nt DNA oligo (5′-GTTTTTAGTTCGGAGGGTC-3′) with a 5′ overhang of 3′-TG-5′ was annealed to

the 3′-end of the RNA transcripts. The Klenow fragment of DNA polymerase I (New England Biolabs) was added to extend the 3′-end of the RNA by incorporation of a single [$\alpha$-$^{32}$P] dATP residue. The 3′-end labeled RNA was then purified by electrophoresis through 8% denaturing polyacrylamide gels. In-line probing was performed as previously described (*Gao et al., 2012*). 3′-end labeled RNA was heated to 75°C and slow cooled to room temperature. One pmol of labelled RNA was allowed to self-cleave at 25°C for 14 hr in in-line buffer (50 mM Tris-HCl pH 8.5, 20 mM MgCl$_2$). The alkaline ladder and RNase T1 digest ladder were generated using an RNase T1 kit (Ambion, Catalog#: AM2283) according to the manufacturer's instructions. All samples were resolved by electrophoresis through 8% denaturing polyacrylamide gels and exposed to phosphorimager screens. The intensities of individual bands were quantified using Semi-Automated Footprinting Analysis (SAFA) software (*Das et al., 2005*). The net signal change in mutant band intensities was normalized to a percentage of wt.

## Protoplast transfection and RNA gel blots

Protoplasts were generated from seed-derived *Arabidopsis thaliana* callus tissue (ecotype Col-0) and were transfected with in vitro transcribed full-length TCV gRNA as described previously (*McCormack and Simon, 2006*). Briefly, 20 µg of uncapped gRNA was transfected into $5 \times 10^6$ cells, and cells incubated for 40 hr in the dark. Total RNA was extracted, subjected to electrophoresis, transferred to a nitrocellulose membrane and probed using [$\gamma$-$^{32}$P] ATP-labeled oligonucleotides complementary to positions 3931–3953, 3869–3883, and 4034–4053 in the 3′UTR. Membranes were exposed to a phosphorimager screen. Experiments were performed in triplicate.

## Preparation of RNA molecules for optical tweezers (OT)

To prepare RNAs for OT, DNA sequences of interest flanked on both sides by 500 bp DNA handle sequences were inserted into pUC19. With the T7 RNA polymerase promoter as part of the forward primer, DNA handle A was PCR amplified from bases 2587 to 401 of pUC19 and DNA handle B was PCR amplified from bases 412 to 892. Handle B was dual-biotinylated at its 5′ end using a dual-biotinylated primer. Digoxigenin (DIG) was incorporated at the 5′ end of handle A using terminal deoxynucleotidyl transferase (Fermentas). In brief, 300 pmol of handle A was incubated with 7500 pmol of DIG-dUTP and 60 U of enzyme at 37°C for 3 hr. Enzymes were removed with phenol:chloroform:isoamyl extraction (25:24:1), pH eight and the excess DIG-dUTP was removed after filtration through a 3K amicon filter column.

RNA fragments for OT analysis were annealed with DIG-handle A and dual biotin-handle B by mixing RNA (1.5 µg) with 3 µg of DNA-handle A and 3 µg of DNA-handles B in 80 µl annealing buffer (80% formamide, 1 mM EDTA, 40 mM PIPES pH 6.3, 0.4 mM NaCl). The mixture was heated to 85°C for 10 min and slow cooled to 65°C for 90 min, 55°C for 90 min and 10°C for 10 min. The annealed products were precipitated with 3 M sodium acetate and 100% ethanol, rinsed with 70% ethanol and dissolved in 100 µl H$_2$O.

## Optical tweezers

Dual laser beam optical tweezers (*Moffitt et al., 2008*) with one optical trap was used for studying the folding properties of the TSS (*Figure 2—figure supplement 1*). Dual lasers were steered by mirrors, refocused using 100x magnification optical lenses, and passed through a flow chamber in the opposite direction, creating an optical trap that holds a 4 µm bead coated with anti-DIG antibody. The position of the trapped bead was monitored by collecting the light scattered off of the bead onto a position sensitive detector. Another 2 µm streptavidin-coated bead was mounted on a micro glass pipette using suction. The position of the micropipette was manipulated by controlling the piezoelectric flexure stage. Force was applied to the RNA molecule held between the two beads by moving the optical trap away from the micropipette. Changes in the extension of the RNA were measured by the relative movement of the trapped bead and the piezoelectric flexure stage. Experiments were performed in 10 mM Tris-HCl pH 7, 250 mM NaCl and with either 10 mM EDTA or 10 mM MgCl$_2$.

## Force-ramping and force-clamping

RNAs were subjected to a force range between 0 pN and 25 pN. In force-ramping, force was applied to a single RNA at a constant rate of 100 nm/s to unfold the molecule, which was followed immediately with refolding by decreasing the force at the same constant rate. In force-clamping, the RNA was held at a constant force for 1.5 to 2 min and the change in extension of the RNA over time was measured. Force was increased by 0.5 pN increments and measurements were retaken to determine the unfolding pathway. In both experiments, any change in extension of the RNA was monitored from the change in the position of the beads; the applied force was determined from the change in the momentum of the laser light passing through the trapped bead. Force and extension were recorded at a rate of 1 k Hz.

## Data analysis

Force versus extension profiles were fitted using the Marko-Siggia Worm Like Chain (WLC) model (*Equation 1*), as the model is more accurate at a low force regime (*Wang et al., 1997*). When fitting the force-extension curves of the DNA/RNA handles, the persistent length was limited between 8 and 10 nm (*Onoa et al., 2003*; *Wen et al., 2007*). Force-clamping data was analyzed by fitting the histogram of the unfolded and folded states with Gaussian distribution to obtain the displacement of the two states. The experimental contour length of the RNA transition was estimated based on the Odjik WLC (*Equation 2*) (*Wang et al., 1997*).

$$F = \left(\frac{k_B T}{L_p}\right)\left[\frac{1}{4\left(1 - \frac{x}{L_0}\right)^2} - \frac{1}{4} + \frac{x}{L_0}\right]$$ (1)

$$x = L_0\left[1 - \frac{1}{2}\left(\frac{k_B T}{F L_p}\right)^{\frac{1}{2}}\right] + \frac{F}{K}$$ (2)

$F$, force; $x$, extension; $L_p$, persistent length; $L_0$, contour length; $K$, elastic modulus (~1000 pN); $k_B T$, Boltzmann's constant times absolute temperature.

## Expression and purification of MBP–RdRp

RdRp was expressed in Escherichia coli and purified as a recombinant protein with maltose binding protein (MBP), as previously described (*Rajendran et al., 2002*). The recombinant p88 plasmid was a gift from P.D. Nagy (University of Kentucky).

## Filter binding

TCV RNAs were in vitro transcribed from PCR-generated templates, dephosphorylated using Antarctic phosphatase (NEB), and radiolabeled with T4 Polynucleotide kinase (NEB) and $\gamma$-$^{32}$P ATP followed by filtration through a G25 (GE) column. Prior to the binding reaction, RNA transcripts were heated to 65°C and allowed to slowly cool to room temperature. Five to 160 pmoles of radiolabeled RNA and 11.4 pmoles of RdRp were incubated together for 30 min at room temperature in binding buffer (50 mM Tris, pH 8.2; 10 mM MgCl$_2$; 10 mM DTT; 10% glycerol) before filtration. GraphPad Prism4 (GraphPad Software) was used to perform one site binding nonlinear regression [equation: Y=Bmax*x/(Kd+x)]. Where Scatchard plots resulted in a nonlinear curve, which indicates a significant non-specific component to binding (see MDV control and mutant m74; *Figure 6*), binding is indicated as 'non-specific' and curves were fitted using a modified one site binding equation (Y=Bmax*x/(Kd+x) + NS*x).

## Molecular dynamics simulations

Simulations were performed using the Amber 14 package with the force field ff14SB that includes the ff99bsc0 and chi.OL3 parameters for RNA (*Case et al., 2014*; *Zgarbová et al., 2011*). Topology and coordinate files were generated with Amber's LEaP module (*Case et al., 2014*). The explicit solvent PME protocol employed was described in our previous studies (*Darden et al., 1993*; *Essmann et al., 1995*; *McCormack et al., 2008*). In brief, RNA was energy minimized and then placed in a TIP3P water solvent box with RNA-charge neutralizing Na$^+$ ions and Na$^+$/Cl$^-$ ion pairs to

simulate 0.1 M salt concentration. Throughout the explicit solvent simulations, periodic boundary conditions were employed and a cutoff of 9 Å for the non-bonded interactions was used. A time step of 2 fs was used with the SHAKE algorithm to constrain all hydrogen bonds in the system. The Berendsen thermostat was used to maintain the temperature and the Berendsen algorithm was used to maintain pressure at 1.0 Pa (*Berendsen et al., 1984*). A multi-step equilibration protocol including heating to 300 K, short dynamics phases, and multiple minimization phases with gradually decreasing harmonic restraints applied to the RNA molecule was followed by an unrestrained production MD simulation of 40 ns.

The previously published model of the TCV TSS (*McCormack et al., 2008*), corresponding to positions $A_{11}$ through $G_{110}$ in *Figure 1A*, was extended at the 5' and 3' ends to the entire TSS118 fragment. This new model (*Figure 1E*) was subjected to explicit solvent molecular dynamics (MD) simulation, which yielded a stable conformation of the 5' end extension and left the 3' extension single-stranded. An MD state at approximately 25 ns was selected as the starting point for the pulling simulations since it captured a representative intermediate state between the start and end conformations, and it is described in the Results section.

Pulling experiments were also simulated in Amber 14, using a Steered Molecular Dynamics Simulations (SMD) protocol. Both the explicit solvent (PME) simulations and implicit solvent (GB) simulation protocols were used as described below. A fragment including C5 through $A_{112}$ (marked in *Figure 1E*), to which we will refer as TSS108, was selected for SMD because it contained all TCV TSS stable interactions and limited the maximum single-stranded chain extension, helping to keep the explicit solvent systems as small as possible (discussed further in the Results section). Heavy atoms $C_5$:O5' and $A_{112}$:O3' were selected to determine the direction of pulling and have the restraints applied to (starting distance between the two selected atoms, target end distance, time interval and spring constant) that simulated pulling at a constant speed (in one simulation) with a spring constant of 6.0 kcal/molÅ. Pulling simulations at 1 Å/ps, 0.5 Å/ps, 0.1 Å/ps and 0.05 Å/ps yielding distinctly different results, as discussed in the Results section. Other PME protocol parameters were the same as in the unrestrained MD (see above).

The implicit solvent simulation methods such as the Generalized Born model (GB) (*Nguyen et al., 2015*; *Tsui and Case, 2000b*), combine an explicit atomic representation for solutes (the 3476 atoms of the TSS TCV fragment in this case) with the solvation free energy approximated as a sum of polar and nonpolar terms. The much smaller size of the simulated system also permitted the use of the fast Amber PMEMD module. The GB-based SMD simulations were performed with constant pulling speeds of 0.01 Å/ps and 0.005 Å/ps, with three variants of intrinsic Born radii parameters: GB (mbondi, igb = 1), GB-HCT (mbondi, offset = 0.13 Å, igb = 1) (*Gaillard and Case, 2011*; *Hawkins et al., 1995*; *Tsui and Case, 2000a*) and the latest GB-neck2 (mbondi3, igb = 8) (*Nguyen et al., 2015*). The Langevin thermostat was employed with a collision frequency of 1.0 ps$^{-1}$ and no cutoff. Simulations were run at 300 K, with a two fs time step, SHAKE, and a Debye-Hückel salt screening concentration of 1.0. No explicit ions were used. The equilibration protocol used for all the GB runs started with energy minimization, followed by heating to 300 K with harmonic restraints of 15 kcal/mol/Å$^2$ applied to the RNA and six MD stages with harmonic restraints gradually lowered to 0.1 kcal/mol/Å$^2$ for the total equilibration time of 2.0 ns. SMD runs that followed the equilibration phase extended TSS108 up to 700 Å in 63.4 and 126.8 ns, depending on the simulated pulling speed. In addition to the SMD runs, the overall stability of the TSS108 model was verified in implicit solvent unrestrained MD simulations with all GB parameter variants at time scales equal to and longer than those employed in SMD.

## Acknowledgements

This work was supported by the National Science Foundation (MCB-1411836) and National Institutes of Health (R21AI117882-01) to AES. Support for this research was also provided by NIH training grant (NIGMS T32GM080201) to MTL. This project has been funded in part with federal funds from the Frederick National Laboratory for Cancer Research, National Institutes of Health, under contract HHSN261200800001E for WKK. This research was supported in part by the Intramural Research Program of the NIH, National Cancer Institute, Center for Cancer Research. The content of this publication does not necessarily reflect the views or policies of the Department of Health and Human Services, nor does mention of trade names, commercial products, or organizations imply

endorsement by the US Government. This study used computational resources and support of the National Cancer Institute's Advanced Biomedical Computing Center.

## Additional information

### Funding

| Funder | Grant reference number | Author |
|---|---|---|
| National Science Foundation | MCB-1411836 | My-Tra Le<br>Feng Gao<br>Megan Y L Young<br>Xuefeng Yuan<br>Anne E Simon |
| National Institutes of Health | R21AI117882-01 | My-Tra Le<br>Feng Gao<br>Anne E Simon |
| National Cancer Institute | Intramural | Wojciech K Kasprzak<br>Taejin Kim<br>Bruce A Shapiro |
| National Institutes of Health | 2T32AI051967-06A1 | Megan Y L Young<br>Anne E Simon |

The funders had no role in study design, data collection and interpretation, or the decision to submit the work for publication.

### Author contributions

M-TL, Conceptualization, Formal analysis, Investigation, Methodology, Writing—original draft; WKK, Conceptualization, Formal analysis, Investigation, Methodology, Writing—original draft, Writing—review and editing; TK, MYLY, XY, Conceptualization, Investigation, Methodology; FG, Conceptualization, Investigation; BAS, Conceptualization, Resources, Formal analysis, Supervision, Project administration; JS, Resources, Supervision, Methodology; AES, Conceptualization, Resources, Formal analysis, Supervision, Funding acquisition, Writing—original draft, Project administration, Writing—review and editing

### Author ORCIDs

Anne E Simon, http://orcid.org/0000-0001-6121-0704

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

## Appendix 1

### Molecular Dynamics and Steered Molecular Dynamics simulations in explicit solvent

The TCV TSS model including all 118 nucleotides of TSS118 was subjected to explicit solvent molecular dynamics (MD) simulations in Amber (*Case et al., 2014*) which indicated a potential placement of the 5′ end residues A7 through A10 in the major groove of $\Psi_3$ (see *Figure 1E*).

A 108 nt-long fragment of the above model between nucleotides C5 and A112 (inclusively) was selected for the Steered Molecular Dynamics (SMD) simulations because it contained all the TCV TSS stable interactions and limited the maximum single-stranded chain extension length, an important factor in explicit solvent simulations (see below). Heavy atoms C5:O5′ and A112:O3′ were selected to determine the direction of pulling (along the top of the TSS 'T') and the distance restraint parameters were applied to them.

OT experiments were performed at pulling speeds that were orders of magnitude slower than our simulations (due to the total computer time required to perform one simulation). Thus, we first assessed the impact of several SMD pulling speeds on the predicted unfolding sequence of events and ran the pulling simulations at the slowest speeds practicable with the explicit, then implicit solvent approaches (see Materials and Methods). The most important finding was that pulling speed influences the orientation of helical regions relative to the direction of pulling, most importantly that of H4b, which, in turn, leads to formation of transient tertiary interactions that change the resistance to pulling and alter the sequence of opening of the key domains in the TSS model. Chen at al. previously observed that the parallel orientation of the helices of a pseudoknot to the direction of pulling (i.e. perpendicular orientation of their base pairs) increases the resistance because a shear force is applied to stems, rather than the unzipping force that opens base pairs one at a time (*Chen et al., 2007*). Our results are consistent with theirs, but additionally indicate that the speed of pulling influences transient orientations of structural elements.

The explicit solvent SMD simulations at a pulling speed of 1.0 Å/ps terminated with helices H5 and H4b remaining partly base-paired, while the 5′ and 3′ ends of the modeled TCV TSS fragment were pulled 700 Å apart, at which distance the chain should be totally single-stranded. The backbone of the single-stranded (opened) portion of the molecule was overstretched, with the straight-line P-to-P atom distances up to 33% above those found in a relaxed A-form helix. An additional 1.0 ns MD simulation was conducted with the ends held 700 Å apart to let the remaining base pairs open and the forces measured at the end to relax to zero. This was a clear indication that a pulling simulation at this speed keeps the TCV RNA away from (ahead of) its equilibrium state. At 0.5 Å/ps pulling speed, the simulation ended with less base paired structure, and only at 0.1 Å/ps pulling speed were all TSS base pairs open at the end of the simulation.

Pulling speed also had an impact on the dynamics of the opening of individual TCV TSS helices. At the highest pulling speed, the H5 helix pivoted toward the horizontal (the direction of pulling), formed tertiary interactions with upstream nucleotides and acted like a pseudoknot, additionally increasing resistance to pulling. At all pulling speeds, after the opening of $\Psi_2$, H4b remained horizontally oriented and formed new transient tertiary interactions between the adenylates in the H4b terminal loop and the uridylates in the single-stranded linker between H4b and H5. These interactions formed a transient H-type pseudoknot and also increased the resistance to pulling. Such transient tertiary interactions that replaced the original pseudoknots or added new ones influenced the order of TSS unfolding, with H4a/$\Psi_3$ opening first, in disagreement with the experimental OT results. Of the seven simulations run at 0.1 Å/ps (two with higher NaCl salt concentrations that did not change the results) only one run yielded a sequence of unfolding events ending with H4a/$\Psi_3$

opening last. These simulations clearly showed dependence of the sequence of unfolding events on pulling speed and pointed to the need for longer SMD runs simulating slower pulling speed. The longest practical explicit solvent simulation times with the pulling speed of 0.05 Å/ps did not improve the results.

To evaluate if slowing down the critical step without slowing the entire simulation would bring the simulations into agreement with the experimental data, a pause in the pulling was introduced after the opening of $\Psi_2$, i.e. when the last hydrogen bond was lost between the H4b and $\Psi_2$. Following this event, the SMD was run for one ns with the TCV TSS ends held at a constant distance (~215 Å) to re-equilibrate the system. During this time, the H4b terminal loop adenylates maintained H-bond contacts with the H4b-H5 linker uridylates, but changed their horizontal orientation to point further away (closer to the perpendicular) from the direction of pulling. After the pause in pulling, the SMD at 0.05 Å/ps was resumed until the full extension of 700 Å was reached. Even though the re-equilibration pause did not lead to immediate full separation of the H4b from the rest of the structure and did not prevent the transient alternative tertiary interaction from forming, the global sequence of structure opening events changed to H4b opening first, H5 opening second and H4a opening last. This was in general agreement with the experimental OT data.

The explicit solvent results indicated the crucial role of the H4b separation from the rest of TSS after $\Psi_2$ opening on the sequence of the TSS unfolding events and dependence of this result on the simulated pulling speed. In order to perform slower pulling speed simulations (i.e. longer SMD runs) in reasonable real time, we employed the implicit solvent simulations (Generalized Born model implemented in Amber). These confirmed the crucial role of the pulling speed in bringing the simulation and experimental results into agreement, and they are described in detail in the text of this study.

