## [Decision Letter]

Thank you for submitting your article "Folding behavior of a T-shaped, ribosome-binding translation enhancer implicated in a wide-spread conformational switch" for consideration by *eLife*. Your article has been favorably evaluated by Wenhui Li (Senior editor) and three reviewers, one of whom, Nahum Sonenberg (Reviewer #1), is a member of our Board of Reviewing Editors.

The reviewers have discussed the reviews with one another and the Reviewing Editor has drafted this decision to help you prepare a revised submission.

Summary:

The manuscript by Le et al., describes an RNA element located in the 3'UTR of turnip crinkle virus that undergoes a conformational switch resulting in a switch from translation to replication. Previous studies have shown that the TSS structure functions as a unit. That its structure is not altered by binding to ribosomes however, it was significantly altered upon RdRp binding. The authors used optical tweezers to understand how this structure folds and unfolds in order to understand how the conformational change is triggered. This work adds to our understanding of the structure of the TSS beyond what has already been published since it includes sequences adjacent that have been shown to be important ribosome binding and stability of the structure. Here the authors show that stem-loop H4A and pseudoknot (PK) 3 form a stable structure that drives cooperative folding of the rest of the TSS structure. Formation of this structure is dependent on 5 upstream adenosines, which can function as a type of molecular switch by either stabilizing the TSS or upon RDRP binding destabilize it. This manuscript provides convincing data for order of folding and unfolding of the TSS using several different approaches. However, this becomes a bit tedious and redundant. The authors also include supporting data on gRNA TCV transcript levels accumulated with mutations in the 5A region and K_d_ measurements of TSS mutants with RDRP that supports their conclusion for how these structures are important for the translation/replication switch. Overall, this is a very nice study that adds to our understanding of the mechanisms for how the TSS structure serves as a molecular switch between translation and replication of the viral RNA. While the detailed study sheds light on the behavior of the TSS, however, more general insights into the folding properties of structured RNAs also come out of this work. This manuscript is generally interesting and would be a good addition to the papers published in *eLife*.

Essential revisions:

1) Subsection “5A is implicated in RdRp binding to a 3’ end fragment”, first paragraph: It seems surprising that a loss of binding of the RDRP to m74 would only reduce transcription by 3.6-fold. Perhaps the authors could address this better. Were there differences in the salt or Mg^2+^ concentrations between the transcription reactions and the binding studies?

2) Many of the figures and labels on the figures are too small to easily read.

3) It is unfortunate that all the SMD simulations were done in the absence of Mg^2+^ when their data clearly shows that Mg^2+^ plays an important role.

4) The authors propose a model where by the RDRP binds to the 5A and downstream regions which disrupts the PK3 and this is what leads to unfolding of the rest of the TSS element. However, addition of an oligo to H4a/PK3 was unable to bind due to the stability of the RNA structure. The question arises as to how the RDRP is binding to this short region that is presumably in accessible to disrupt H4a/PK3, since the authors suggest that some of the adenosines interact with the major groove of PK3.

---

## [Author Response]

Essential revisions:

1) Subsection “5A is implicated in RdRp binding to a 3’ end fragment”, first paragraph: It seems surprising that a loss of binding of the RDRP to m74 would only reduce transcription by 3.6-fold. Perhaps the authors could address this better. Were there differences in the salt or Mg^2+^ concentrations between the transcription reactions and the binding studies?

The transcription reactions contained 100 mM potassium glutamate, which was not present in the binding reactions. Other salt and magnesium concentrations were identical. It is known that the 3’ terminal hairpin in conjunction with upstream hairpin H4 is sufficient to confer a low level of transcription by the RdRp. The following sentence was added to this section and the new reference was added to references:

“The ability of the RdRp to continue transcribing low levels of complementary strands in the absence of 5A binding is likely due to the presence of the 3’ terminal Pr hairpin, which in combination with upstream hairpin H4 can promote a low level of transcription (Sun and Simon, 2006).”

2) Many of the figures and labels on the figures are too small to easily read.

We have enlarged the fonts in most of the figures.

3) It is unfortunate that all the SMD simulations were done in the absence of Mg^2+^ when their data clearly shows that Mg^2+^ plays an important role.

As we stated in the Molecular Dynamics section (perhaps not sufficiently clearly) we cannot perform reliable MD or SMD simulations including Mg^2+^ ions because we have no experimental data on Mg^2+^ ion placement in TCV TSS. Ab initio prediction of proper Magnesium ions placement and determination of their influence on the TCV TSS structure is not a feasible alternative because it is currently impossible to perform microsecond time scale simulations that would be necessary to achieve proper Mg^2+^ coordination.

We can only speculate, partly based on the observed impact of extended persistence of tertiary interactions involving H4b in some SMD simulations (as described in MD simulations of TSS unfolding section and in the Supplementary Information), that Psi2-Mg^2+^ interactions could alter the order of TSS building blocks opening, perhaps leading to the H4b/Psi2 and H5 opening in parallel, coupled by magnesium-mediated tertiary interactions (that would be observed as one OT rip), while the Mg-strengthened H4a/Psi3 could still offer proportionally more resistance to pulling and open last.

4) The authors propose a model where by the RDRP binds to the 5A and downstream regions which disrupts the PK3 and this is what leads to unfolding of the rest of the TSS element. However, addition of an oligo to H4a/PK3 was unable to bind due to the stability of the RNA structure. The question arises as to how the RDRP is binding to this short region that is presumably in accessible to disrupt H4a/PK3, since the authors suggest that some of the adenosines interact with the major groove of PK3.

Binding of an oligonucleotide requires formation of particular hydrogen bonds that are not necessarily the hydrogen bonds (and ionic bonds) that are involved in protein binding. Furthermore, the RdRp may be recognizing the adenylates within the structure of H4a/Psi3, which would be very different from accessibility to unpaired (or breathing) nucleotides that are required for the oligonucleotide.